# Deciphering Cell Lineage Gene Regulatory Network via MTGRN

## Abstract

Gene regulatory network (GRN) inference is crucial for cell fate decision, as it outlines the regulations between genes, which direct cell differentiation. Although there have been some work to infer cell lineage GRN, they fail to capture the continuous nature of the differentiation process as they group cells by cell type or cluster and infer GRN in a discrete manner. In this paper, we hypothesize GRN can forecast future gene expression based on history information and transform the inference process into a multivariate time series forecasting problem, linking cells at different time to learn temporal dynamics and inferring GRN in a continuous process. We introduce MTGRN, a transformer-based model that only takes single cell data as input to infer the cell lineage GRN by forecasting gene expression. MTGRN consists of temporal blocks and spatial blocks, effectively captures the connections between cells along their developmental trajectories and leverages prior knowledge to elucidate regulatory interactions among genes. It significantly outperforms six other methods across five datasets, demonstrating superior performance even compared to multimodal approaches. Based on the inferred GRN, MTGRN pinpoints three crucial genes associated with the development of mouse embryonic stem cells and depicts the activity changes of these genes during cellular differentiation. Beyond this, MTGRN is capable of conducting perturbation experiments on key genes and accurately modeling the change of cell identity following the knockout of the Gata1 in mouse hematopoietic stem cells.

## 1 Introduction

Organism develops from a single fertilized egg, which undergoes differentiation to produce various specialized cell types necessary for maintaining essential life activities. Although all cells in a multicellular organism share the same genome, the complex regulatory relationships between genes lead to selective gene expression in different cell types, resulting in the translation of distinct proteins that drive cells toward specific functional fates (Keller, 2005). As shown in Figure 1 (a), gene expression is primarily regulated by transcription factors (TFs), which are proteins that bind to distal cis-regulatory elements (CREs) on DNA and work in concert with cofactors and other proteins to recruit and stabilize the RNA polymerase complex. This, in turn, modulates the transcriptional rate of target genes (TGs), either positively or negatively. In addition to TFs, other elements such as splicing factors, microRNAs, and metabolites can also regulate gene expression. **However, in this paper, we focus exclusively on the regulatory interactions between TFs and TGs.**

Gene regulatory network (GRN) serves as computational models to explain the regulation of gene expression, which are mathematically defined as graphs (Dai et al., 2024). It's noteworthy that TFs are gene products, the regulatory relationship between a TF and its TG can be understood as the relationship between the gene which encodes the TF (TFG) and the TG. Figure 1 (b) demonstrates the topology of GRN, the nodes of the GRN are composed of genes, with some genes are TFGs, while the remaining genes are TGs. The edges of the GRN represent regulatory interactions between genes, with directionality pointing from TFGs to TGs. Inferring cell lineage GRN has been a longstanding goal in biology. Since GRN can be represented as graph, uncovering its topological features and dynamic changes is crucial for understanding how cellular identities are established and maintained, which holds significant potential for engineering cell fates and disease prevention. Originally, GRN was typically constructed by inferring gene co-expression from bulk transcriptomic data (Margolin et al., 2006; Langfelder & Horvath, 2008; Huynh-Thu et al., 2010). However, such

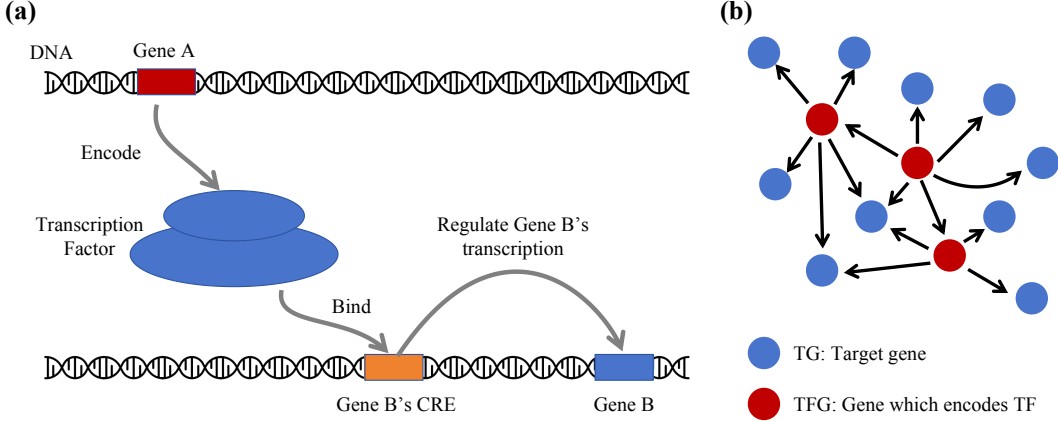

Figure 1: The regulatory processes between genes and the mathematical representation of GRN. (a) Gene A regulates the transcription of Gene B through the TF it encodes. This TF binds to the CRE of Gene B and influences the recruitment of RNA polymerase. (b) GRN can be mathematically represented as a directed graph, where nodes represent genes and directed edges denote regulatory relationships. TFG regulates the expression of TG through the TF it encodes, while the TFG itself can also act as a target gene.

statistically-driven approaches require a large number of samples, which severely limit their practical applicability. Single-cell RNA sequencing (scRNA-seq) (Tang et al., 2009) has addressed some of these limitations by enabling high-throughput sequencing of tens of thousands of cells, allowing researchers to infer GRN at relatively low cost. However, scRNA-seq data do not directly capture many underlying regulatory mechanisms, such as the TF protein abundance and DNA binding events. As a result, some researchers incorporated additional regulatory information, such as single-cell ATAC sequencing (scATAC-seq) (Buenrostro et al., 2015) data into their models, leading to the development of multimodal GRN inference methods (Kartha et al., 2022; Kamimoto et al., 2023; Wang et al., 2023a). Despite the promising results of these approaches, several challenges remain: 1) Due to technical limitations and cost constraints, it is difficult to obtain samples with multiple modalities in practical settings, making large-scale application of these methods challenging. 2) These methods often split cells into groups based on cell type or cluster and infer GRN separately for each group. It disregards the continuous nature of cellular development and fails to capture the dynamic changes during cell differentiation.

To address these challenges, we introduce MTGRN, a transformer-based model comprising both temporal and spatial blocks. MTGRN only utilizes scRNA-seq data as input and frames GRN inference as a multivariate time-series (MTS) forecasting task by assigning pseudotime to each cell along the developmental trajectory. Incorporating prior knowledge from databases and experimental results, MTGRN predicts gene expression at the next $N$ time points based on the gene expression of cells from the previous $M$ time points, learning an attention matrix in the spatial block. This matrix captures the attention scores between genes, which we interpret as the inferred GRN along the cell lineage. Our contributions are summarized as follows:

- Our method is the first work to frame GRN inference task in a MTS forecasting problem. Unlike previous methods that group cells and infer GRN in a discrete manner, we link cells along the developmental trajectory using pseudotime and infer GRN in a continuous process through MTS prediction, enabling a deeper understanding of the dynamic changes in cellular development.

- Our method achieves superior performance across five datasets compared to other approaches.

- Our method is capable of identifying key TFGs that have been validated by biological experiments.

- Our method enables in silico perturbation experiments by propagating the effects through the inferred GRN, simulating the cellular response to perturbations.

## 2 RELATED WORK

Over the past few decades, numerous methods for GRN inference have emerged. We categorize these methods into unimodal and multimodal approaches, which we will introduce in the following paragraphs. Additionally, we will present the trajectory inference (TI) methods employed in MTGRN for pseudotime calculation.

**Unimodal GRN inference.** These methods only take scRNA-seq data as input, aiming to infer regulatory relationships between genes based solely on gene expression information. One of the most commonly used approaches is Weighted Gene Co-expression Network Analysis (WGCNA) (Langfelder & Horvath, 2008), which identifies modules of co-expressed genes by calculating pairwise correlations across the transcriptome. However, since WGCNA built undirected networks based purely on co-expression, it lacks the ability to infer causal regulatory relationships, leading to many false positive associations. To overcome these issues, methods like GENIE3 (Huynh-Thu et al., 2010) and its more efficient variant GRNBoost2 (Moerman et al., 2019) incorporated prior knowledge of regulatory activity to predict target gene expression based only on TF expression, thereby reducing the number of potential interactions. In addition, NetREX (Wang et al., 2018) reconfigured the prior network to find the optimal topology that explains the observed expression data, enhancing the inference of regulatory networks and CEFCON (Wang et al., 2023b) utilized graph neural network to refine the prior network and obtained the ultimate GRN. DTGN (Guo & Xiao, 2024) proposed a framework for identifying phenotype-specific transcription factors and pathways by constructing dynamic transcriptional regulatory networks using time-series gene expression data and a graph autoencoder model. However, relying exclusively on transcriptomic data will generate false positives, as other regulatory mechanisms, such as chromatin accessibility, are often neglected. These shortcomings limit the accuracy of network inference.

**Multimodal GRN inference.** Recent advances in scATAC-seq, which provides insights into chromatin accessibility, identifying regulatory elements and transcription factor binding sites, have allowed for the refinement of GRN reconstruction in a multimodal manner. CellOracle (Kamimoto et al., 2023) integrated scATAC-seq data to simulate changes in cell identity following TF perturbations. By combining computational perturbation with GRN modeling, CellOracle enabled a systematic interpretation of context-dependent TF functions in regulating cell identity. Dictys (Wang et al., 2023a) constructed a initial TF binding network using scATAC-seq data and refined it by modeling transcriptional dynamics with an Ornstein-Uhlenbeck (OU) process to capture biological variability. While multimodal approaches have delivered promising results in GRN inference tasks, most of them clustered cells and then inferred a GRN for each cluster. This discretized strategy disrupts the continuous nature of cellular development.

**Trajectory inference.** TI methods assign to every cell a so-called pseudotime, a numeric value in arbitrary units which measures how far a particular cell is within a dynamic process of interest. By ordering the cells according to this pseudotime, it becomes possible to define the different transition stages through which a cell progresses during its dynamic process. Currently, there are numerous TI methods, such as PAGA (Wolf et al., 2019), Slingshot (Street et al., 2018), and Waterfall (Shin et al., 2015). These methods take either a single snapshot of a mixture of cells at different stages or a set of samples collected at multiple time points as input. According to these single cell data, they aim to order the cells based on an underlying dynamic process that accounts for the heterogeneity observed in the sample. The goal of TI methods is to automatically reconstruct the cellular dynamic process by arranging individual cells sampled and profiled from the process along a trajectory. This trajectory is then used to identify the various stages in the dynamic process and reveal their interrelationships.

## 3 METHODOLOGY

We present MTGRN, a transformer-based model to infer cell lineage GRN, which utilizes self-attention to obtain the correlations between genes in differentiation context. We summarize the framework in Figure 2, where MTGRN consists of three key modules:

1. ***Trajectory Inference Module*** transforms the raw data into time series data by assigning pseudotime to each cell.

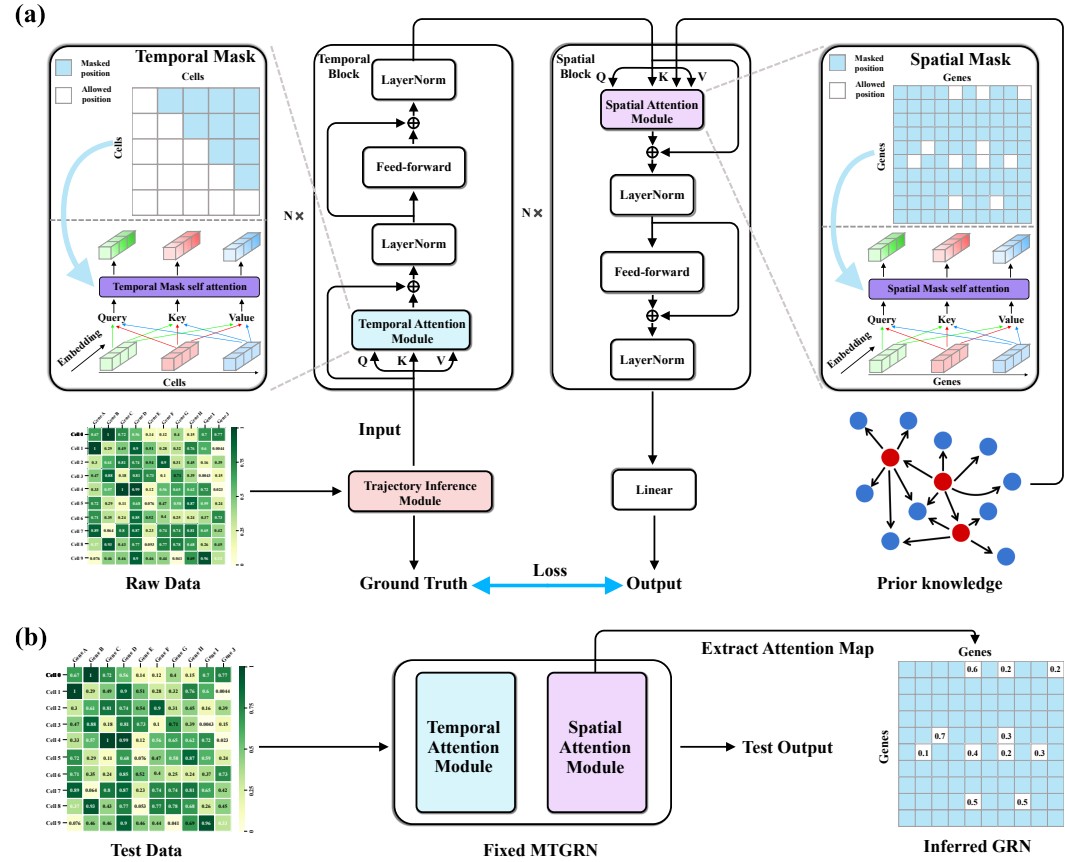

Figure 2: Overview of MTGRN's training and inference process. (a) The data processed by TI method will serve as input to MTGRN. The input will sequentially pass through temporal block and spatial block to learn gene regulatory relationships during the differentiation process. (b) Once the model is trained, we fix the parameters of MTGRN and feed test data into the network. By extracting the attention matrix from the spatial block, we derive the inferred GRN.

2. **Temporal Attention Module** designs a upper triangular mask to learn the intercellular correlations during differentiation process.

3. **Spatial Attention Module** creates a spatial mask based on the prior knowledge to learn the regulatory relationships between genes.

We will introduce them in the following subsections.

## 3.1 TRAJECTORY INFERENCE MODULE

To address the limitation of previous methods, which require cells to be split into groups by cell type or cluster, we employ TI method to transform the original single-cell data into time-series data and infer GRN in a MTS prediction manner. Specifically, as illustrated in Figure 3, our raw data is a gene expression matrix $\mathbf{E} \in \mathbb{R}^{C \times G}$ obtained by scRNA-seq, where $C$ and $G$ represents the number of cells and genes, $\mathbf{E}_{ij}$ indicates the expression value of gene $j$ within cell $i$. To endow the raw data with temporal information, we employ the Slingshot algorithm (Street et al., 2018), one of TI methods, to calculate a pseudotime for each cell. Then we sort the cells in chronological order, thus transforming $\mathbf{E}$ into a time series data $\mathbf{E}^{(t)} \in \mathbb{R}^{T \times G}$, where $T$ is the number of time steps. It's worth noting that $T$ is equal to $C$, as each cell corresponds to a pseudotime on differentiation trajectory. In this way, our MTS data $\mathbf{E}^{(t)} = [\mathbf{e}_1, \dots, \mathbf{e}_T]^T$ possesses the temporal relationships of genes during the differentiation process as we map each cell to a pseudotime, where $\mathbf{e}_i \in \mathbb{R}^G$

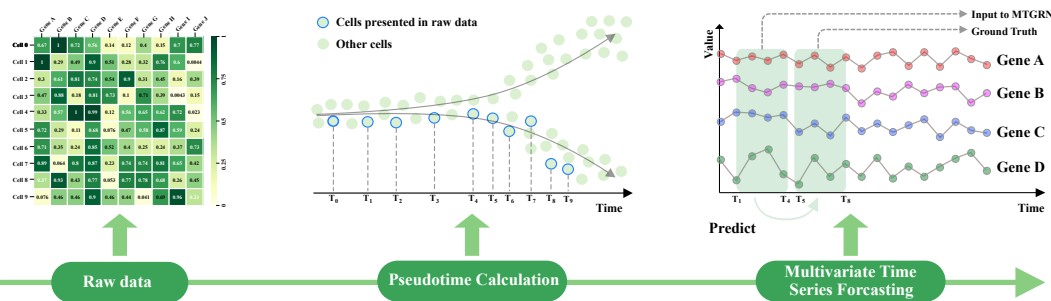

Figure 3: Utilizing TI methods to convert single-cell data into time-series data, where the gene expression profiles from the previous $W$ time points are used as input, and the subsequent $M$ time points serve as ground truth.

represents the gene expression at the $i$-th time step. After that, we split $E^{(t)}$ into dataset $\mathcal{D} = \{(\mathbf{X}_i, \mathbf{Y}_i) \,|\, i = 1, \ldots, N\}$, where $\mathbf{X}_i$ are input samples, $\mathbf{Y}_i$ are the ground truth associated to each sample and $N$ is the number of samples. Each sample $\mathbf{X}_i = [\mathbf{e}_i, \ldots, \mathbf{e}_{i+W-1}]^T \in \mathbb{R}^{W \times G}$ and $\mathbf{Y}_i = [\mathbf{e}_{i+W}, \ldots, \mathbf{e}_{i+W+M-1}]^T \in \mathbb{R}^{M \times G}$ is the gene expression in $W$ observation windows and $M$ prediction windows respectively.

***Since GRN regulates gene expression, we assume it can forecast the subsequent gene expression within $M$ windows ($\mathbf{Y}_i$) by observing the expression within proceeding $W$ windows ($\mathbf{X}_i$).***

Therefore, the inference process can be formulated as a MTS forecasting problem and our goal is to learn a mapping function $f_{GRN}$:

$$f_{GRN}(\mathbf{e}_{t-W+1}, \ldots, \mathbf{e}_t) = (\mathbf{e}_{t+1}, \ldots, \mathbf{e}_{t+M}) \tag{1}$$

### 3.2 Temporal Attention Module

As shown in Figure 2 (a), we will embed each value from the output of ***Trajectory Inference Module***, that is $\mathbf{X} \in \mathbb{R}^{W \times G}$ to $\mathbf{X} \in \mathbb{R}^{W \times G \times d_{model}}$ and transpose it to $\mathbf{X}_{input} \in \mathbb{R}^{G \times W \times d_{model}}$, where $d_{model}$ is the embedding dimension. $\mathbf{X}_{input}$ will be input into ***Temporal Attention Module***, where we construct a unique temporal mask $\mathbf{M}^t \in \mathbb{R}^{W \times W}$ for calculation of attention:

$$\mathbf{M}^t_{ij} = \begin{cases} -\inf, & \text{if } j > i \\ 1, & \text{if } j \leq i \end{cases} \tag{2}$$

$\mathbf{M}^t$ constraints cell at a given time point can't attend to information at subsequent cells. This trick is in line with the principles of cell differentiation, as a cell at specific time can only know the gene information from its ancestors but not its descendants, because they haven't been born yet at this point. Thus, the calculation of temporal attention is modified as follows:

$$\text{TemporalAttention}(Q, K, V) = \text{softmax}\left(\frac{QK^T}{\sqrt{d_k}} \odot \mathbf{M}^t\right) V \tag{3}$$

where $d_k = \dfrac{d_{model}}{h}$, $h$ is the number of heads in multi-head self-attention and $\odot$ represents hadamard product.

### 3.3 Spatial Attention Module

As shown in Figure 2 (a), the output of temporal block $X_{output} \in \mathbb{R}^{G \times W \times d_{model}}$ will be transposed to $\hat{X}_{output} \in \mathbb{R}^{W \times G \times d_{model}}$, which will be input into ***Spatial attention module*** alongside prior knowledge. The prior knowledge is a highly comprehensive gene interaction network proposed in NicheNet (Browaeys et al., 2020), which is a collection of gene regulatory interactions from over 50 public data sources of mouse and human, we filter the genes involved in the prior knowledge to

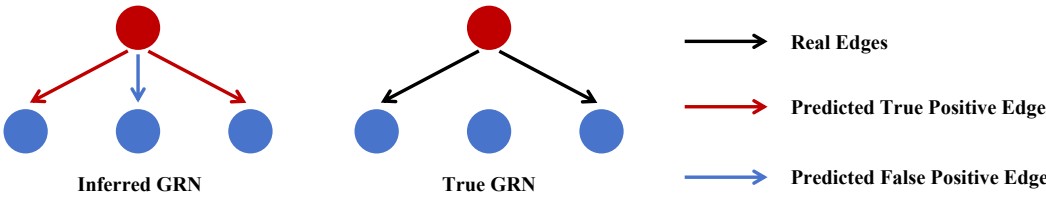

Figure 4: Metric calculation involves treating the GRN inference as a binary classification problem for edges, where each dataset has a corresponding true GRN. We assess whether the predicted edges are present in the true GRN.

those present in the single-cell input, ensuring the prior knowledge only focuses on the relationships among the $G$ genes. Since prior knowledge can be represented as a graph, we use its adjacency matrix $\mathbf{P} \in \mathbb{R}^{G \times G}$ as a mask in the computation of self attention:

$$\mathbf{M}^s_{ij} = \begin{cases} -\inf, & \text{if} \quad \mathbf{P}_{ij} = 0 \\ 1, & \text{if} \quad \mathbf{P}_{ij} = 1 \end{cases} \quad (4)$$

$\mathbf{M}^s$ ensures the attention is computed between genes that are known to have regulation in prior knowledge, rather than focusing on unrelated genes. Thus, the calculation of spatial attention can be formulated as follows:

$$\text{SpatialAttention}(Q, K, V) = \text{softmax}\left(\frac{QK^T}{\sqrt{d_k}} \odot \mathbf{M}^s\right)V \quad (5)$$

Ultimately, the output of spatial block will go through a linear layer to obtain the predicted gene expression $\widehat{\mathbf{Y}}_i \in \mathbb{R}^{M \times G}$. We use mean square error (MSE) as our loss function:

$$\mathcal{L} = MSE(\widehat{\mathbf{Y}}_i, \mathbf{Y}_i) \quad (6)$$

### 3.4 GRN INFERENCE

Once the model is trained, we fix the parameters of MTGRN and use the test data to compute the attention matrix in the ***Spatial Attention Module***, as illustrated in Figure 2 (b). The values in the attention matrix $\mathbf{H} \in \mathbb{R}^{G \times G}$ represent the regulatory scores between genes. It is important to note that the regulatory scores in the masked regions are zero, as no regulatory edges exist between those gene pairs in the prior knowledge. We then multiply the regulatory scores by the degree of TFG in the network, rank the scores in descending order, and select the top $K$ edges to form the inferred GRN, where $K$ corresponds to the number of edges in the true gene regulatory network.

## 4 EXPERIMENT SETTINGS

**Datasets.** We assessed the quality of GRN inferred by our model on five cell lineage datasets: human mature hepatocytes (hHep) (Camp et al., 2017), embryonic stem cells (mESC) (Hayashi et al., 2018), and three mouse hematopoietic stem cell lineages (mHSC-E, mHSC-GM, mHSC-L) (Hayashi et al., 2018). For each lineage, we used the ground truth network provided in Pratapa et al. (2020) to evaluate the accuracy of the inferred GRN.

**Baselines.** We compared MTGRN with: (1) GENIE3 (Huynh-Thu et al., 2010), GRNBoost2 (Moerman et al., 2019), NetREX (Wang et al., 2018) and CEFCON (Wang et al., 2023b), four methods that use scRNA-seq data combined with prior knowledge; (2) CellOracle (Kamimoto et al., 2023), a multimodal approach that integrates scRNA-seq and scATAC-seq data and (3) Random, where edges are randomly selected to infer the GRN.

**Metrics.** We computed the areas under the precision-recall (AUPRC) and receiver operating characteristic (AUROC) curves, using the edges in the true GRN as ground truth and the ranked edges from each method as predictions. As illustrated in Figure 4, the comparison between the predicted

Table 1: Performance comparison with six methods on five benchmark datasets.

| | hHep | | | mESC | | | mHSC-E | | | mHSC-GM | | | mHSC-L | | |
|---|---|---|---|---|---|---|---|---|---|---|---|---|---|---|---|
| | AUROC | AUPRC | F1 | AUROC | AUPRC | F1 | AUROC | AUPRC | F1 | AUROC | AUPRC | F1 | AUROC | AUPRC | F1 |
| GENIE3 | 0.481 | 0.084 | 0.167 | 0.531 | 0.168 | 0.275 | 0.350 | 0.019 | 0.026 | 0.419 | 0.072 | 0.138 | 0.486 | 0.183 | 0.322 |
| GRNBoost2 | 0.578 | 0.077 | 0.101 | 0.548 | 0.143 | 0.217 | 0.385 | 0.007 | 0.018 | 0.450 | 0.068 | 0.122 | 0.515 | 0.181 | 0.297 |
| NetREX | 0.575 | 0.096 | 0.110 | 0.509 | 0.191 | 0.299 | 0.515 | 0.091 | 0.162 | 0.466 | 0.120 | 0.205 | 0.509 | 0.182 | 0.300 |
| CEFCON | 0.465 | 0.218 | 0.391 | 0.494 | 0.291 | 0.448 | **0.552** | 0.379 | 0.502 | 0.623 | 0.634 | 0.686 | **0.653** | 0.659 | 0.675 |
| Celloracle | 0.527 | 0.341 | 0.478 | 0.502 | 0.204 | 0.336 | 0.465 | 0.268 | 0.460 | 0.462 | 0.270 | 0.457 | 0.564 | 0.278 | 0.365 |
| Random | 0.481 | 0.032 | 0.068 | 0.513 | 0.111 | 0.198 | 0.500 | 0.083 | 0.155 | 0.495 | 0.090 | 0.167 | 0.518 | 0.135 | 0.227 |
| **MTGRN (ours)** | **0.664** | **0.639** | **0.651** | **0.713** | **0.748** | **0.694** | 0.485 | **0.429** | **0.635** | **0.765** | **0.849** | **0.859** | 0.583 | **0.791** | **0.879** |

and true GRN frames the task as a binary classification problem for the edges. AUPRC and AU-ROC serve as key evaluation metrics for binary classification and we also calculated the F1 score to evaluate the accuracy of inferred GRN.

**Reproducibility.** We used the Adam optimizer with a warmup strategy to increase the learning rate from 0 to 1e-4, followed by CosineAnnealingLR scheduler for further adjustments. The $d_{model}$ and number of heads of Transformer were set to 128 and 4 respectively. We trained our model on a 80G Nvidia A100 GPU with 20 epochs and implemented an early stopping strategy to prevent overfitting, where the patience was set to 3.

## 5 RESULTS

### 5.1 MTGRN OUTPERFORMS OTHER METHODS

Table 2: Statistical analysis of the label distribution for each dataset.

| | hHep | mESC | mHSC-E | mHSC-GM | mHSC-L |
|---|---|---|---|---|---|
| Genes | 805 | 774 | 961 | 949 | 639 |
| Positive edges | 2019 | 8085 | 5394 | 6508 | 4705 |
| Possible edges | 647220 | 598302 | 922560 | 899652 | 407682 |
| Proportion | 0.003 | 0.014 | 0.006 | 0.007 | 0.012 |

To evaluate the accuracy of MTGRN in GRN inference problem, we compared it against six methods across five developmental datasets. The quantitative results are shown in Table 1. In terms of AUROC, MTGRN achieved the highest performance on four out of the six datasets. Although its performance on the mHSC-E and mHSC-L datasets were slightly below that of CEFCON, MTGRN consistently outperformed other methods in AUPRC and F1 scores across all six datasets. For instance, on hHep dataset, MTGRN achieved AUPRC of 0.639, surpassing the second-best method CellOracle (0.341) by 0.298 and showing an improvement of F1 score by 0.173. When compared with CellOracle, the multimodal GRN inference method, we found that despite using only single cell data, MTGRN outperformed CellOracle across all metrics in five datasets, showcasing its superior performance. Furthermore, we observed that most methods exhibited low AUPRC. To have a deeper insight into this phenomenon, we analyzed the label distribution in each dataset, as summarized in Table 2. **Genes** is the number of genes in the input scRNA-seq data. **Possible edges** refers to the total number of potential regulatory edges between genes, which was calculated as all possible pairwise combinations but exclude the self-loops. It can be represented as $G \times (G - 1)$, where $G$ is the number of genes. **Positive edges** denotes the actual edges in the true GRN, and **Proportion** reflects the ratio of true regulatory edges within the total gene pair search space. From Table 2, We observed that the **Proportion** in all datasets were approximately 0.01, suggesting there is a severe label imbalance in these datasets (positive labels are significantly fewer than the negative labels). In this case, AUPRC becomes a more suitable metric than AUROC for evaluating model performance (Davis & Goadrich, 2006). As a result, despite other models showing poor performance in AUPRC, MTGRN still achieved an AUPRC higher than 0.6 (exclude the mHSC-E dataset), and even reached AUPRC over 0.85 on the mHSC-GM and mHSC-L datasets, demonstrating the superior performance of our method.

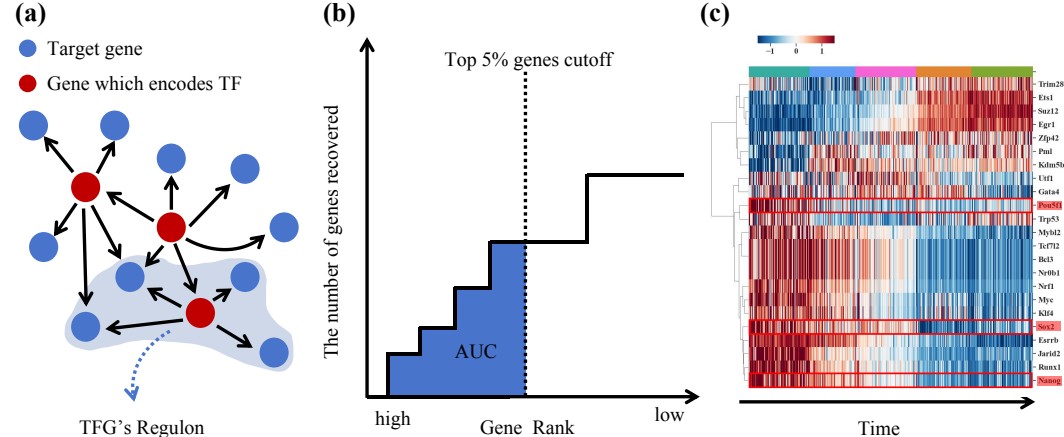

Figure 5: Workflow for identifying key TFGs using the AUCELL algorithm. (a) The TFG's regulon is consisted of the TFG and its target genes. (b) AUCELL calculates the TFG's activity score in a given cell by assessing the proportion of regulon genes that rank among the top highly expressed genes. (c) A heatmap of z-scores for each TFG's activity score, showing the dynamic changes in TFG activity throughout different stages of cell differentiation.

## 5.2 MTGRN IDENTIFIES KEY TFGS

Identifying and understanding the key TFGs that drive cell differentiation is crucial for uncovering the fundamental biological mechanisms of life. These TFGs not only play a central role in understanding cell fate decisions in basic research but also have significant potential applications in stem cell therapy, disease research. For example, recognizing specific TFGs can deepen our understanding of the molecular basis of diseases, particularly those caused by aberrant gene regulation, such as cancer or developmental disorders (Anderson et al., 2023; Walsh et al., 2017). Additionally, by manipulating these critical TFGs, scientists can precisely control cell differentiation in experimental settings, paving the way for the development of innovative cell therapies and organ regeneration techniques.

We chose mouse embryonic stem cells, the mESC dataset shown in Table 1 to evaluate our model's ability in identifying key TFGs. Mouse embryonic stem cells exhibit high pluripotency, allowing them to differentiate into nearly all cell types within the embryo, including neurons, cardiomyocytes, and hepatocytes. This remarkable differentiation potential makes mESC an ideal model for studying developmental biology and the underlying mechanisms of cellular differentiation. We trained MTGRN on the mESC dataset and used the AUCELL algorithm proposed by SCENIC (Aibar et al., 2017) to score the activity of each TFG's regulon in each cell and identify the key TFGs based on the activity score.

Specifically, AUCELL takes a gene set as input and outputs the gene set activity score for each cell. As shown in Figure 5 (a), these gene sets correspond to regulons, which are composed of TFGs and their predicted target genes. AUCELL calculates the enrichment of each regulon by determining the area under the recovery curve (AUC) based on the ranking of all genes in a given cell, where genes are ranked according to their expression levels. In brief, as illustrated in Figure 5 (b), the x-axis represents the ranking of all genes by expression level (genes with identical expression values, such as 0, are randomly ordered), and the y-axis represents the number of genes recovered from the input TFG regulon. AUCELL uses the AUC to assess whether a critical subset of the input regulon is enriched at the top of the ranking for each cell (by default, selecting the top 5% of genes). This way, the AUC score reflects the proportion of genes from the TFG regulon that are highly expressed in each cell, with higher scores indicating greater TFG activity in that cell. The output of this step is a matrix containing AUC scores for each TFG regulon across all cells.

After the calculation of the activity score for all TFGs involved in the GRN inferred by MTGRN, we divided the cells into two groups based on developmental pseudotime, representing early and late developmental stages. For each TFG, we conducted differential analysis between the early-stage

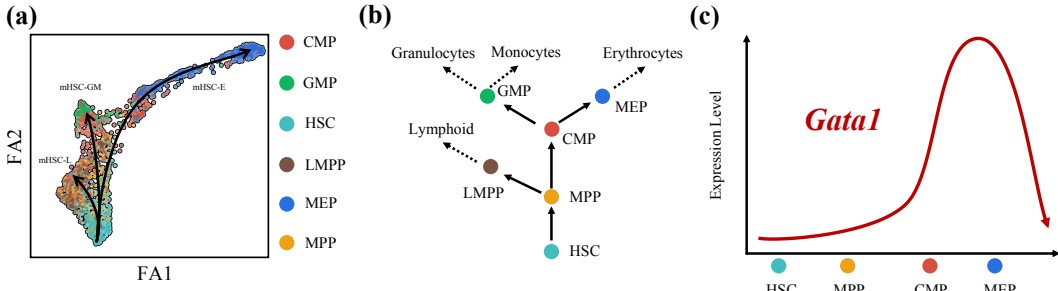

Figure 6: Overview of the mHSC dataset. (a) Force-directed visualization of the three developmental lineages. (b) In the mHSC dataset, all development originates from HSC and ultimately leads to the differentiation into three distinct cell types. (c) In the erythrocytes lineage (mHSC-E), the expression level of Gata1 changes dynamically throughout the development process.

and late-stage cell groups and calculated the p-value for each TFG. TFGs with p-values less than 0.01 were filtered out and considered key TFGs showing significant differences during lineage development. As shown in Figure 5 (c), we identified a total of 23 differential TFGs and found that three TFGs, namely Nanog, Sox2 and Pou5f1 (Oct3/4), which are well-known pluripotency factors for mESC development (Wang et al., 2012). It had been previously reported that the three TFGs were mutually regulated by one another, forming cross-regulated feedforward loops (Almeida et al., 2021). We then calculated the z-scores of the activity scores for the 23 identified key TFGs and presented them in a heatmap. As shown in Figure 5 (c), the activity scores for Nanog, Sox2, and Pou5f1 are high during the early stages and gradually decline as the cells develop. This pattern aligns with previous studies, which report that Pou5f1, Nanog, and Sox2 are highly expressed in undifferentiated embryonic stem cells and the three TFGs are crucial for maintaining the pluripotency of stem cells (Masui et al., 2007). All these results suggested that MTGRN can accurately identify the key TFGs determining cell fates.

## 5.3 MTGRN Performs In Silico Gene Perturbation

MTGRN not only can identify key TFGs involved in cell development but also enable in silico perturbation experiments. Unlike traditional laboratory perturbations, in silico experiments allow for the rapid and cost-effective exploration of potential changes within large gene regulatory networks, eliminating the need for extensive experimental resources or time-consuming wet-lab procedures. By simulating gene knockouts or overexpression scenarios, we can predict how gene regulatory networks will respond to various perturbations, thus accelerating our understanding of critical regulatory factors.

We selected the mouse hematopoietic stem cell (mHSC) dataset from Table 1 for analysis. Figure 6 (a) illustrates a force-directed (FA) visualization of the three differentiation lineages (i.e., mHSC-E, mHSC-GM, mHSC-L). The cell type annotations were obtained from Nestorowa et al. (2016): HSC refers to hematopoietic stem cells, MPP to multipotent progenitors, LMPP to lymphoid multipotent progenitors, CMP to common myeloid progenitors, MEP to megakaryocyte-erythrocyte progenitors, and GMP to granulocyte-monocyte progenitors. Figure 6 (b) provides a detailed representation of the three differentiation lineages of mHSC. It starts from HSC and eventually leading to the formation of lymphoid cells, erythrocytes, granulocytes and monocytes. Gata1 is known to orchestrate significant changes in the expression of genes throughout the erythrocytes differentiation process, driving critical steps required for the proliferation and differentiation of erythrocytes progenitors. As shown in Figure 6 (c), Gata1 expression is initiated at the CMP stage during early erythrocytes commitment and gradually decreases as erythrocytes mature (Moriguchi & Yamamoto, 2014). Given that Gata1 is a well-established TFG, we aim to assess whether MTGRN can accurately simulate the effects of knocking out this gene.

As shown in Figure 7 (a), because GRN can be abstracted as a graph, the effects of perturbing TFGs can propagate through its neighboring nodes. In Figure 7 (b), we simulate such a perturbation by setting the expression value of a gene to zero, resulting in a change denoted as $\triangle X$, which

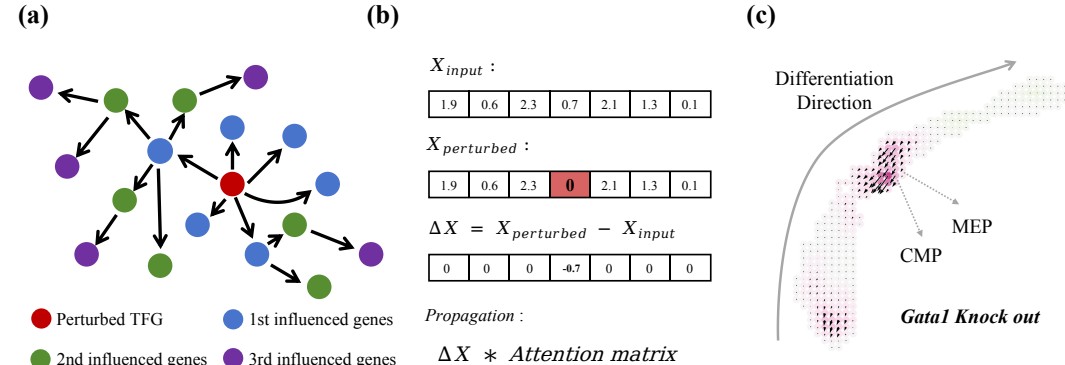

Figure 7: In silico perturbation workflow. (a) A GRN can be represented as a graph, where perturbing a specific gene causes the effects to propagate through its neighboring nodes. (b) After perturbing a gene, the resulting gene expression changes are propagated by multiplying the differences with the attention map from the MTGRN spatial block. (c) In the mHSC-E dataset, perturbing Gata1 results in transition vectors pointing back toward HSC, opposite to the original differentiation direction.

represents the difference of gene expression after perturbation. The propagation of this perturbation is modeled by multiplying $\triangle X$ with the attention matrix $\mathbf{H} \in \mathbb{R}^{,G,\times,G}$ obtained from spatial block in MTGRN (section 3.4), where the attention scores in $\mathbf{H}$ indicate the weights to which the influence of this perturbation will be propagated to neighboring nodes. Once the propagation is complete, each cell will have a perturbed gene expression value. Using these values, we calculated a probability vector for each cell, indicating the likelihood of transitioning toward different cells as a result of the gene perturbation, the calculation process is same as Kamimoto et al. (2023). As illustrated in Figure 7(c), after setting the Gata1 gene expression to zero and propagating the influence, we computed the transition vectors for each cell. It can be observed that these vectors predominantly point toward HSC, suggesting that cells fail to proceed with erythrocytes differentiation when Gata1 is knocked out, which is consistent with previous report that Gata1 promotes the differentiation of HSC into erythrocytes (Moriguchi & Yamamoto, 2014). Additionally, the transition vectors are more pronounced in areas where MEP and CMP cells are clustered, indicating that these cells are more significantly affected by the Gata1 perturbation, consistent with earlier reports that Gata1 expression begins in CMP cells (Moriguchi & Yamamoto, 2014). These experiments demonstrate that the GRN inferred by MTGRN can accurately predict the cellular changes after key TFG perturbations, showcasing the model's strong performance in GRN inference.

## 6 CONCLUSION

We introduced MTGRN, the first approach that formulated the GRN inference process as a MTS problem. Our method inferred GRN in a continuous process compared to approaches that group cells by type or cluster. MTGRN uses only scRNA-seq data as input, combined with prior knowledge to learn cell lineage GRNs. MTGRN outperformed six other inference methods (one for multimodal inference method) across five datasets. Furthermore, in the mESC dataset, MTGRN successfully identified three key TFGs previously reported and the activity scores of theses TFG all exhibited clear trends of change (initially high then low or initially low then high), with Nanog, Sox2, and Pou5f1 (Oct3/4) all showing a trend from high to low. This is consistent with previous reports that the three TFGs are crucial for maintaining the pluripotency of stem cells, in other words, they are highly expressed in the initial embryonic stem cells and have lower expression in differentiated cells. In addition, in the mHSC-E dataset, after knocking out Gata1 and propagating the influence through our inferred GRN, we validated that the knock out of Gata1 significantly inhibited the differentiation of CMP and MEP cells, which is consistent with the finding that Gata1 promotes the differentiation of these two cell types into erythrocytes. All of results demonstrate the accuracy of our inferred GRN and we think MTGRN offers a novel approach to GRN inference.

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
