# OpenReview forum: "Deciphering Cell Lineage Gene Regulatory Network via MTGRN"
_ICLR.cc/2025/Conference — Submitted to ICLR 2025_

### Official Review · Reviewer_wyvQ · 2024-10-24

**Soundness:** 1
**Presentation:** 2
**Contribution:** 2
**Rating:** 5
**Confidence:** 5

**Summary:**

This paper proposes MTGRN, a transformer-based model that performs GRN inference from scRNA-seq data. The method first orders cells via a trajectory inference algorithm and then treats the problem as a time series forecasting task. Two attention modules are proposed that capture connections between cells and genes. The method is compared against several baselines on 5 datasets.

**Strengths:**

- The paper is well-written and easy to understand. The proposed method is clear and straightforward.
- Several datasets and baselines are considered to establish the improved performance of the proposed method.

**Weaknesses:**

This paper has a few weaknesses which I detail below. Addressing these would strengthen the paper in my opinion.

- The proposed method incorporates prior knowledge in the form of a known GRN (NicheNet) to limit the space of possible regulatory links to those that are known. This defeats the purpose of the algorithm as the validation essentially compares two established GRNs— NicheNet and the ground truth used in the experiments—likely resulting in a significant overlap. It is unclear why this approach is considered superior against baselines which do not use such prior information but consider all GxG connections as possible (e.g., GENIE3, GRNBoost2). The substantial improvement in scores might be attributed to this unfair advantage.
- There is no experiment to show that the top K edges selected are not simply derived by the most expressed genes/TFs (which are likely to be the ones enriched in the corresponding cell lineages).
- Several variables such as Q, K, V are not defined in the paper nor supplement. It is not clear how the input to the TemporalAttention module $X_{\text{input}}$ of shape $G\times W\times d$ is transformed into queries, keys to give a matrix of length $W$. Furthermore, Q, K, V in the Spatial attention module seem to have different meaning and dimension than Q, K, V defined prior.
- The use of attention for GRN inference from scRNA-seq data has been explored before [1] which limits the novelty of this paper in my view.

[1] https://academic.oup.com/bioinformatics/article/39/4/btad165/7099621

**Questions:**

Authors rely on a trajectory inference method to order cells by differentiation time, which could introduce additional hyperparameters/variance. Why not use time-series scRNA-seq datasets where the time points are given rather than learned?

---

> ### Author Response · Authors · 2024-11-14
>
> We appreciate the reviewer’s high-quality comments. Below, we will address each of the questions raised.
> 1. In response to the reviewer's question that NicheNet and the ground truth network have a significant overlap. It is a very important point, we will discuss below. First, we proposed a method named **Random** in Table 1 for baseline comparison. In **Random**, we randomly selected the top-k edges (where k is equal to the number of edges in the ground truth) from the prior knowledge as the inferred GRN. As for results, we can find that the AUROC for **Random** across five datasets is around 0.5,  indicating there is no significant overlap between the prior knowledge and the ground truth. Second, we analyzed the overlap between the edges provided by NicheNet and the edges in ground truth. The result is presented below:
>    - Prior Knowledge: 5318181 edges
>    - mESC ground truth network: 24557 edges (overlap: 16695)
>    - mHSC-E ground truth network: 24726 edges (overlap: 14732)
>    - mHSC-GM ground truth network: 16198 edges (overlap: 9818)
>    - mHSC-L ground truth network: 4705 edges (overlap: 2494)
>
>     we can find that NicheNet includes 5318181 edges, while the ground truth edge counts in our datasets are 24557, 24726, 16198, and 4705, with intersections of 16695, 14732, 9818, and 2494 edges, respectively. In each dataset, the effective regulatory edges provided by NicheNet account for only 0.31%, 0.27%, 0.18%, and 0.04% of the entire prior knowledge. This indicates that NicheNet provides a coarse prior network containing considerable noise, including potential false-positive regulatory relationships from databases or experiments. This is also why the AUROC for random is around 0.5 in Table 1. From above, there is no significant overlap between NicheNet and the ground truth network, which is why CEFCON [1] also uses it as prior knowledge. MTGRN should identify approximately 10,000 true regulatory edges from a noisy network of over 5 million edges, making this task challenging. For the second question about GENIE3 and GRNBoost2, we chose to compare our model with them for two key reasons. First, these methods are considered classic in GRN inference, so comparing against them allowed us to benchmark our model's performance against established standards. Second, this comparison emphasize incorporating prior knowledge can enhance GRN inference. Except for GENIE3 and GRNBoost2, the other three baseline methods (NetREX, CEFCON, and Celloracle) all rely on prior networks, in comparison with these three methods, our model consistently demonstrated significant performance gains, showing the effectiveness of our method. **If reviewer has any recommendations for models suitable for comparison, we would be glad to compare our model with them as well**
> 2. In selecting the top k edges, we did not incorporate any gene enrichment information. Using the mESC dataset as an example, MTGRN needs to select the top 24557 edges  out of a total of 5318181 edges available in the NicheNet network, the ground truth edges is a tiny fraction of the entire search space, which includes numerous false-positive regulatory interactions that could interfere with network inference. Due to this noise, some unrelated genes may appear more enriched in the prior network than actual regulatory factors. Because MTGRN will predict the gene expression for future M time points based on the former N time points, if an unrelated gene highly enriched in the prior network, it will not effectively fit the expression in the M future cells, so during attention map computation, our model will assigns lower scores to such genes, meaning that even if a gene appears highly enriched, it will still receive a low score if it does not contribute to accurate predictions (beacause they are not the true regulatory factors for the target genes). We can refer to the method **Random** in Table 1 for a clearer explanation. Since **Random** randomly selects edges from the prior network, it is more likely to pick enriched genes. However, it achieved an AUROC close to 0.5 across all five datasets. If our model’s edge selection were also based on gene enrichment, our scores would be similar to those of **Random**. Instead, MTGRN’s accuracy across all five datasets is significantly higher than **Random**, demonstrating that our selection of the top-k edges is not based on gene enrichment.
>
> [1] Wang P, Wen X, Li H, et al. Deciphering driver regulators of cell fate decisions from single-cell transcriptomics data with CEFCON[J]. Nature Communications, 2023, 14(1): 8459.
>
> **Due to the character limit, we will continue addressing the remaining questions in the next comment.**

---

> > ### Comment · Reviewer_wyvQ · 2024-11-15
> >
> > I thank the authors for their reply. Please find my comments below.
> >
> > 1. Did you also restrict the edges selected by GENIE3, GRNBoost2, etc to those from NicheNet? It seems to me like this would be a more appropriate comparison if you wish to maintain the prior graph. What are the results of your method if you remove NicheNet entirely from the pipeline? Also, don't the overlap values you present mean that there are edges in the ground truth graphs that are not part of NicheNet, therefore never picked by your method? This may complicate the interpretation of the scores. It is also unclear what the overlap is between the ground truth and the prior information in the other baselines (NetREX, CEFCON, and Celloracle) that use such prior information.
> > 2. It would help to show a quantile plot of the TFs (or targets) selected by your method, when measured against a ranking of TFs or targets by total counts/expression value. If these genes belong to, say, the top 1% most expressed genes, then I do not see an advantage of running this method compared to simply taking the top expressed genes. It is unclear to me why authors say "Random is more likely to predicted enriched genes".

---

> > > ### Author Response · Authors · 2024-11-16
> > >
> > > We sincerely thank reviewer for their response. we will answer the questions below.
> > > 1. Regarding the first question, *"Did you also restrict the edges selected by GENIE3, GRNBoost2, etc., to those from NicheNet?"*, in the submitted version, we did not intersect the edges predicted by GENIE3 and GRNBoost2 with those in NicheNet. We agree with the reviewer's suggestion that performing such an intersection would provide a more appropriate comparison. Consequently, we conducted this experiment, and the AUROC scores of these two methods after intersecting with NicheNet on the five datasets are as follows (the numbers in parentheses indicate the original scores.):
> > >
> > >      |             |      hHep     |      mESC     |     mHSC-E    |    mHSC-GM    |     mHSC-L    |
> > >      |:-----------:|:-------------:|:-------------:|:-------------:|:-------------:|:-------------:|
> > >      |  GRNBoost2  | 0.589 (0.578) | 0.562 (0.548) | 0.413 (0.385) | 0.473 (0.450) | 0.534 (0.515) |
> > >      |    GENIE3   | 0.503 (0.481) | 0.552 (0.531) | 0.378 (0.350) | 0.442 (0.419) | 0.498 (0.486) |
> > >      | MTGRN(ours) |     0.664     |     0.713     |     0.485     |     0.765     |     0.583     |
> > >
> > >    It can be observed that the AUROC scores of GENIE3 and GRNBoost2 improved after intersecting with NicheNet, but the improvement is not significant. This is because the prior knowledge provided by NicheNet contains many false-positive edges. As a result, intersecting with NicheNet still retains these incorrectly predicted edges. This further validates that the prior network we used is a very coarse network.
> > >
> > > 2. Regarding the second question, *"What are the results of your method if you remove NicheNet entirely from the pipeline?"* The AUROC scores of our method on the five datasets after removing the prior knowledge are as follows (the numbers in parentheses indicate the original scores):
> > >
> > >      |       |      hHep     |      mESC     |     mHSC-E    |    mHSC-GM    |     mHSC-L    |
> > >      |:-----:|:-------------:|:-------------:|:-------------:|:-------------:|:-------------:|
> > >      | MTGRN | 0.631 (0.664) | 0.692 (0.713) | 0.431 (0.485) | 0.738 (0.765) | 0.557 (0.583) |
> > >
> > >    We found that without the prior knowledge network, the performance of our model decreased, but the decline was not significant. Moreover, our model still achieved the best results on the hHep, mESC, and mHSC-GM datasets, even surpassing methods like NetREX, Celloracle, and CEFCON that utilize prior knowledge.
> > >
> > > 3. Regarding the third question, *"don't the overlap values you present mean that there are edges in the ground truth graphs that are not part of NicheNet, therefore never picked by your method?"* This is indeed correct. Since the prior knowledge network does not contain all the edges in the ground truth, the GRN inferred by the model will not include those missing edges. This is a limitation inherent to methods that rely on prior knowledge networks, including Celloracle and CEFCON. However, we want to emphasize that this phenomenon only affects the model's performance since there will always be some ground truth edges that the model cannot predict, which in turn lowers its overall performance. Taken together, the use of the NicheNet network does not significantly enhance the model's performance.
> > >
> > > 4. For the fourth question, *"It is also unclear what the overlap is between the ground truth and the prior information in the other baselines (NetREX, CEFCON, and Celloracle) that use such prior information,"* we want to emphasize that the prior networks used for comparison with NetREX and CEFCON are both derived from NicheNet. However, for Celloracle, as it constructs its base GRN using ATAC-seq data for various species, we calculated Celloracle's scores using its own prior network. Next, we will analyze and report the overlap between Celloracle's prior network and the ground truth, with the results provided below:
> > >
> > >      |   |       mESC      |      mHSC-E     |     mHSC-GM    |     mHSC-L    |
> > >      |:-:|:---------------:|:---------------:|:--------------:|:-------------:|
> > >      |   | 5490099 (31868) | 5490099 (13966) | 5490099 (9498) | 5490099(3178) |
> > >
> > >    It can be observed that the number of edges in Celloracle's prior network and the overlap with the ground truth network are similar to those of NicheNet, demonstrating the rationale of using NicheNet as the prior.
> > >
> > > **Due to the character limit, we will continue addressing the remaining questions in the next comment.**

---

> > > ### Author Response · Authors · 2024-11-16
> > >
> > > 5. Regarding the fifth question, *"If these genes belong to, say, the top 1% most expressed genes, then I do not see an advantage of running this method compared to simply taking the top expressed genes,"* we completely agree with the reviewer's point. To address this, we conducted an experiment using mESC dataset. We summed the expression values of each gene across all cells and selected the top 20 most highly expressed genes. Additionally, we extracted the top 20 TFs with the most target genes in the inferred GRN by MTGRN. The results are as follows:
> > >
> > > **the top 20 genes by experession**
> > >
> > > | Gene       | Expression Value |
> > > |------------|------------------|
> > > | Hsp90aa1   | 2794.064339      |
> > > | Actg1      | 2762.883397      |
> > > | Hspa8      | 2756.692880      |
> > > | Trim28     | 2537.954558      |
> > > | Sparc      | 2494.004587      |
> > > | Hspa5      | 2354.852788      |
> > > | Lama1      | 2247.966072      |
> > > | Lamc1      | 2234.615596      |
> > > | Prdx1      | 2224.608706      |
> > > | Calr       | 2196.692511      |
> > > | Hsp90b1    | 2143.477250      |
> > > | Lamb1      | 2028.298558      |
> > > | Calm1      | 2026.405005      |
> > > | Pdia3      | 2015.682167      |
> > > | P4hb       | 2003.225678      |
> > > | Serpinh1   | 1944.835849      |
> > > | Bsg        | 1899.764351      |
> > > | Surf4      | 1887.487196      |
> > > | Myl6       | 1871.722739      |
> > > | Lrpap1     | 1862.228798      |
> > >
> > >
> > > **the top 20 genes with most target gene in inferred GRN**
> > >
> > > | Gene   | Target Genes |
> > > |--------|--------------|
> > > | Myc    | 762          |
> > > | Nanog  | 744          |
> > > | Runx1  | 728          |
> > > | Nrf1   | 718          |
> > > | Pou5f1 | 711          |
> > > | Utf1   | 706          |
> > > | Klf4   | 703          |
> > > | Ets1   | 695          |
> > > | Trim28 | 681          |
> > > | Suz12  | 669          |
> > > | Egr1   | 633          |
> > > | Sox2   | 610          |
> > > | Kdm5b  | 601          |
> > > | Tcf7l2 | 557          |
> > > | Pml    | 529          |
> > > | Esrrb  | 522          |
> > > | Trp53  | 511          |
> > > | Mybl2  | 503          |
> > > | Zfp42  | 465          |
> > > | Nfya   | 431          |
> > >
> > > We are excited to find that the key genes identified through the inferred GRN are entirely different from those obtained using gene expression. Furthermore, the genes we identified, such as *Pou5f1*, *Nanog*, and *Sox2*, are well-documented as being associated with mouse embryonic stem cell development. This demonstrates that our model can accurately infer GRNs and uncover relevant key genes that are distinct from those identified by merely analyzing gene expression.
> > >
> > > 6. Regarding the sixth question, *"It is unclear to me why authors say 'Random is more likely to predict enriched genes,'"* we would like to clarify this point. **Random** randomly chooses edges from the prior network. If a gene is enriched in the network, meaning it has a significantly higher number of connections compared to others, the edges connected to this gene are more likely to be randomly selected.
> > >
> > > Moreover, based on feedback from other reviewers, we have uploaded the experimental results of MTGRN identifying dynamic GRNs in the supplementary material. Additionally, we included the predicted gene expression metrics such as spearmanR or pearsonR in the supplementary material. Hoping reviewer could refer to this content for a comprehensive evaluation of the value of our work. Thank you very much!
> > >
> > > We are very grateful for your valuable suggestions and hope our responses clarify any concerns. We greatly value this opportunity to discuss our work. If there are any further questions, please feel free to ask, and we will be happy to provide further details. We hope this discussion will help improve the score of our paper, and thank you again!

---

> > > > ### Comment · Reviewer_wyvQ · 2024-11-18
> > > >
> > > > I thank the authors for the additional experiments. I have thus raised my score to a 5. However, I think the contribution of the paper is limited to justify a higher score.

---

> ### Author Response · Authors · 2024-11-14
>
> 3. In response to reviewer’s comment that several variables are not defined in the paper. We sincerely apologize for the oversight in explaining some of the symbols, we will provide a detailed explanation below. $X_{\text{input}} \in \mathbb{R}^{G \times W \times d}$ will go through three independent linear transformation to obtain Q, K, V, they are all have shape of $G \times W \times d$. To calculate the attention matrix, we use the formula:$\frac{Q K^\top}{\sqrt{d}}$, the result is an attention matrix with dimensions $G \times W \times W$. As for the second quesiton, in **Section 3.3** of the paper, we mention that
> > the output of temporal block $X_{output} \in \mathbb{R}^{G \times W \times d_{model}}$ will be transposed to $X_{output} \in \mathbb{R}^{W \times G \times d_{model}}$, which will be input into Spatial attention module alongside prior knowledge
>
>    The difference in dimensions between the spatial attention module and the temporal attention module, as mentioned by the reviewer, is due to a transposition we applied to the output of the temporal module. We switched the positions of G and W, allowing us to compute the attention between genes in the spatial attention module. Consequently, the attention map in the spatial attention module has dimensions of $G \times G$.
> 4. In response to reviewer’s comment that already have work using transformers for GRN inference, we want to emphasize that MTGRN’s contribution is not applying transformer to infer GRN (**we did not mention this as a contribution in our introduction**). Our key contribution is reframing GRN inference as a multivariate time series forecasting problem. MTGRN transforms single-cell data into time sequence data using trajectory inference and then infers GRNs through a time-series prediction approach. The proposed time and spatial modules are designed to learn Granger causality [1] in the cell development process. In the time module, each cell can only attend to cells from prior time points, allowing each gene to observe all genes in preceding cells. However, only a small subset of genes influences the expression of a given target gene. To address this, we use the spatial module to filter out gene pairs without regulatory relationships in the prior knowledge, ensuring that each gene’s expression is inferred based only on historical expression data of genes with known interactions in prior network. Transformer is merely a tool we use for GRN inference, our main contribution lies in restructuring the GRN inference task as a new paradigm of multivariate time series prediction problem.
> 5. In response to question that why not use time-series scRNA-seq datasets where the time points are given rather than learned? The datasets we use for benchmaring come from BEELINE [2]. For example, in the mESC dataset, each cell is named in a format like “RamDA_mESC_00h_C01,” which indicates its collection time, here at 0 hours. The mESC data were collected at five distinct time points (0, 12, 24, 48, and 72 hours). As for why we don’t use these specific time points directly, there are three main reasons. First, in each collection time point, multiple cells are sequenced to obtain their gene expression data, so these cells will have the same time label (e.g., 00h), which prevents us from establishing a precise temporal order among them. Second, we think cells collected at the same time still have an inherent time order. By assigning a pseudotime to cells collected at same time point, we can establish a more accurate order within each time point. Third, previous methods such as CellOracle [3] inferred GRN by clustering cells of the same type and then constructed a GRN for each cluster. However, we believe that even within the same cell type, there is an inherent developmental progression. Therefore, we use trajectory inference to assign pseudotimes to cells, allowing us to capture this developmental order more effectively.
>
> We are very grateful for your valuable suggestions and hope our responses clarify any concerns. We greatly value this opportunity to discuss our work. If there are any further questions, please feel free to ask, and we will be happy to provide further details. We hope this discussion will help improve the score of our paper, and thank you again!
>
> [1] Shojaie A, Fox E B. Granger causality: A review and recent advances[J]. Annual Review of Statistics and Its Application, 2022, 9(1): 289-319.
>
> [2] Pratapa A, Jalihal A P, Law J N, et al. Benchmarking algorithms for gene regulatory network inference from single-cell transcriptomic data[J]. Nature methods, 2020, 17(2): 147-154.
>
> [3] Kamimoto K, Stringa B, Hoffmann C M, et al. Dissecting cell identity via network inference and in silico gene perturbation[J]. Nature, 2023, 614(7949): 742-751.

---

### Official Review · Reviewer_rGeJ · 2024-10-29

**Soundness:** 2
**Presentation:** 2
**Contribution:** 2
**Rating:** 6
**Confidence:** 4

**Summary:**

Genes are known to work together in specific pathways and form gene regulatory networks (GRNs). GRNs govern cell differentiation in both normal and disease conditions and identifying GRNs is crucial to understand developmental processes. This is an important area of research and the authors propose a multivariate time series forecasting problem where given single cell RNA-seq data and prior information and gene interaction, an attention based model is used comprising both temporal and spatial information to predict the gene expression in future time points. Representing [genes x cells] matrix as a [genes x times] matrix and using causal attention blocks is a smart idea to formulate a time-series prediction problem. Adding spatial attention using prior interaction networks is interesting as it tells the model to pay attention to those genes that are known to interact. The proposed approach shows that GRN prediction results is better than the benchmark methods on all except mHSC-E and mHSC-L cell types. Overall this is a promising approach and should help with generating more ideas.

**Strengths:**

1. The use of causal attention in time series problem to predict future gene expression is smart. Using spatial attention from prior gene regulatory networks also interesting.
2. Choosing embryonic stem cells show that the GRNs can be used to study cell differentiation
3. Perturbation of gene expression results on Gata1 is very interesting and that the results correspond to the past finding that Gat1 mediates significant changes in the expression of genes throughout the erythrocytes differentiation process shows the method has promise.

**Weaknesses:**

1. To prove the model and the approach is robust the authors could show perturbation of other known TFs and show how does it affect the GRNs.
2. The authors focus on the GRN prediction, and did not show metrics on the gene expression prediction itself.
3. While it is interesting to show that the model can confirm previously found important genes/transcription factors such as Gata1, it does not show any new networks or interactions between TFs and TGs even with some lower confidence. Validation of predicted GRNs that contain previously unknown genes can be done with knockout experiments and could be shown.
4. Authors could cite. Constructing the dynamic transcriptional regulatory networks to identify phenotype-specific transcription regulators which also focuses on. learning temporal representations of gene.

**Questions:**

1. How did the predicted gene expression metrics such as spearmanR or pearsonR look like?
2. Does the model understand genes that are co-regulated by multiple transcription factors? For e.g. https://www.nature.com/articles/s41467-019-11905-3 paper shows that EGr1 recruits Tet1 during development and upon neuronal activity. What happens to the gene expression of a target gene that is regulated by multiple TFs when the expression of just one TF is perturbed and the second TF is undisturbed?
3. Does the model show any new gene-gene interactions?

---

> ### Author Response · Authors · 2024-11-15
>
> We thank the reviewer for their recognition of our paper and for the valuable feedback provided. We sincerely appreciate these insights! Below, we will address each of the questions raised.
> 1. In response to reviewer's comments that how did the predicted gene expression metrics such as spearmanR or pearsonR look like? We conducted the relevant experiments and included the results in the supplementary material. The results demonstrate that MTGRN accurately models gene expression, with Spearman R and Pearson R scores both exceeding 0.98.
> 2. In response to reviewer's comments that does the model understand genes that are co-regulated by multiple transcription factors? We sincerely thank the reviewer for raising such an excellent question, which will guide the next steps in optimizing MTGRN. Currently, we are sorry that MTGRN has not delved deeply into this aspect, but we wiil try this in the future.
> 3. In response to reviewer's comments that Does the model show any new gene-gene interactions? Actually, MTGRN is capable of handling this task. Once trained, during the inference stage, we can obtain the embedding for each gene before calculating the attention map. By simply computing the similarity between gene embeddings, we can predict new gene interactions. After obtaining the gene embeddings, of course more complex methods can be employed to predict new gene interactions, we have proposed a straightforward approach to demonstrate that MTGRN is capable of handling this task.
>
> **We want to emphasize that we have also included the results of MTGRN identifying dynamic GRNs in the supplementary material.** In fact, MTGRN is able to infer dynamic GRN, as we assign each cell a pseudotime and use time-series prediction to infer the GRN. During inference, we can segment cells along the differentiation trajectory into different time segments (e.g., grouping every $n$ consecutive cells into one segment). By inputting cells from each segment into MTGRN, we can obtain the GRN for that specific time segment. For the entire differentiation trajectory, assuming we generate $m$ time segment, we can apply Gaussian smoothing to the regulatory edge scores across these $m$ GRNs to construct a dynamic GRN. This approach is similar to Dictys [1].
>
> We are very grateful for your valuable suggestions and hope our responses clarify any concerns. We greatly value this opportunity to discuss our work. If there are any further questions, please feel free to ask, and we will be happy to provide further details. We hope this discussion will help improve the score of our paper, and indeed we will cite the extraordinary paper reviewer mentioned in comments, thank you again!
>
> [1] Wang L, Trasanidis N, Wu T, et al. Dictys: dynamic gene regulatory network dissects developmental continuum with single-cell multiomics[J]. Nature Methods, 2023, 20(9): 1368-1378.

---

> > ### Comment · Reviewer_rGeJ · 2024-11-21
> >
> > Thank you for clarifying the results on dynamic network results. As pointed out the method is similar to the Dictys paper. Would have loved to see some more validation of the predicted networks.
> > Also looking forward to perturbation of experiments on co-regulated genes in future!

---

> > > ### Author Response · Authors · 2024-11-25
> > >
> > > We sincerely appreciate the valuable feedback provided by the reviewer. Over the past few days, we have followed the Dictys GitHub repository’s notebook [1] to generate a comparative analysis between Dictys and our model.
> > >
> > > Firstly, **we would like to emphasize that Dictys requires multimodal data**, specifically scATAC-seq data, whereas our method only relies on scRNA-seq data. For the mESC dataset, we collected the corresponding ATAC-seq data in a prior study [2] and used it as input for Dictys. **The NCBI GEO accession number for the mESC ATAC-seq data is GSE159623**. The performance metrics of Dictys and our model on the mESC dataset are summarized as follows:
> > > |       | MTGRN | Dictys |
> > > |:-----:|:-----:|:------:|
> > > | AUROC | 0.713 |  0.603 |
> > > | AUPRC | 0.748 |  0.432 |
> > > | F1    | 0.694 |     0.507   |
> > >
> > > From the results, we observed that despite Dictys utilizing multiomics data (i.e., ATAC-seq data), its performance in GRN inference accuracy remains inferior to our model. We attribute this to the fact that Dictys requires an initial TF-target network derived from ATAC-seq data using external software tools, which likely introduces significant cumulative errors.
> > >
> > > We sincerely hope this comparative experiment addresses the reviewer’s concerns. Finding the corresponding ATAC-seq data was extremely challenging, and we dedicated considerable time and effort to conducting this experiment. We kindly ask the reviewer to comprehensively evaluate the value of our work and hope will improve our score. Thank you once again!
> > >
> > > [1] https://github.com/pinellolab/dictys/blob/master/doc/tutorials/full-multiome/notebooks/3-static-inference.ipynb
> > >
> > > [2] Zhu Y, Yu J, Gu J, et al. Relaxed 3D genome conformation facilitates the pluripotent to totipotent-like state transition in embryonic stem cells[J]. Nucleic acids research, 2021, 49(21): 12167-12177.

---

> ### Author Response · Authors · 2024-11-25
>
> We carefully read the article “Constructing the dynamic transcriptional regulatory networks to identify phenotype-specific transcription regulators” which focuses on gene temporal dynamics. This approach highlights how dynamic characteristics of transcription regulators can reveal phenotype-specific transcription factors (TFs) and pathways, addressing limitations in static TRN models. The framework’s use of graph autoencoders and statistical methods for identifying dynamic interactions let me learn a lot
> and we cited it in the ***Related Work*** section in our latest submitted version.

---

> > ### Comment · Reviewer_rGeJ · 2024-11-25
> >
> > Thank you for comparing the metrics with the reference method and adding to the reference!

---

> > > ### Author Response · Authors · 2024-11-25
> > >
> > > We are very pleased to have addressed your concerns and hope that the additional experiments we conducted will allow the reviewer to evaluate our work more comprehensively. We hope this discussion will help improve the score of our paper and thank you again!

---

> > > > ### Comment · Reviewer_rGeJ · 2024-11-26
> > > >
> > > > I appreciate the sincere effort the authors put in to address comments from all reviewers. The network predictions look robust compared to the first version. However I am keeping my score same (6). The current contribution of the paper does not demand a higher score in my opinion.

---

### Official Review · Reviewer_G2id · 2024-11-03

**Soundness:** 3
**Presentation:** 3
**Contribution:** 2
**Rating:** 6
**Confidence:** 4

**Summary:**

The paper presents a novel MTGRN model for inferring cell lineage GRNs, which employs transformer architecture to analyze single-cell data. The combination of temporal and spatial blocks effectively captures the intricate relationships between cells and their developmental trajectories. The authors provide compelling empirical evidence of MTGRN's superiority, outperforming six other methods across five datasets, including multimodal approaches. The perturbation experiments further demonstrate the model's practical utility in understanding cellular identity dynamics.

**Strengths:**

The paper presents a novel perspective on gene regulatory network (GRN) inference by framing it as a multivariate time series forecasting problem. This innovative approach allows for capturing the continuous dynamics of cell differentiation, which is a significant advancement over traditional methods that rely on discrete clustering.

The author describes the fundamental algorithm well, and they seem to give all relevant information to understand and reproduce their algorithm.

The proposed method is relative better than previous methods, which is not lack of significance.

**Weaknesses:**

The paper mentions that the advantage of the algorithm lies in dynamic network inference; however, the experimental analysis is based on data from different cell lines rather than dynamic or developmental data, which undermines the convincingness of the experimental results.
Moreover, the authors did not compare their method with latest state-of-the-art methods.

**Questions:**

1. To make their results more convincing, they should compare their method with more latest state-of-the-art methods.
2. The complexity of the MTGRN model may pose challenges for replication and application in other studies. A more thorough explanation of the model's architecture and hyperparameter settings would help researchers understand and implement the model effectively.
3. They should incorporate dynamic gene expression data to infer dynamic networks.

---

> ### Author Response · Authors · 2024-11-14
>
> We sincerely appreciate the reviewer’s thoughtful comments. we will address each of the questions below.
> 1. In response to reviewer's comments that the experimental analysis is based on data from different cell lines rather than dynamic or developmental data. We want to emphasize that **all the data we use is dynamic developmental data**, reviewer could refer to BEELINE [1] for more context. It provide detailed descriptions of the five datasets and we summarize the main details in BEELINE below :
>   - mHSC dataset is sourced from [2] and includes 1,656 hematopoietic stem and progenitor cells (HSPCs), which can be divided into three lineages: mHSC-E (erythroid lineage), mHSC-L (lymphoid lineage), and mHSC-GM (myeloid lineage). Each lineage includes all cells in the progression from the starting cell to the endpoint cell.
>   - mESC dataset is sourced from [3], contains scRNA-seq results from 421 cells that track the development of mouse embryonic stem cells into primitive endoderm cells. These cells were collected at five distinct time points: 0, 12, 24, 48, and 72 hours.
>   - hHep dataset is from [4] and includes scRNA-seq results from an experiment where induced pluripotent stem cells (iPSCs) were differentiated into hepatocyte-like cells. This dataset includes 425 scRNA-seq measurements taken at various time points: day 0, day 6, day 8, day 14 and day 21.
>
>   In summary, all five datasets we use involve dynamic data related to cell development.
>
> 2. In response to reviewer's comments that we did not compare with latest state-of-the-art methods. In fact, the comparison methods CEFCON [5] and CellOracle [6] are GRN inference models published in Nature Communications and Nature in 2023, which, in our view, represent the best-performing models in the GRN inference filed. **If reviewer has any recommendations for models suitable for comparison, we would be glad to compare our model with them as well.**
> 3. In response to reviewer's comments that a more thorough explanation of the model's architecture and hyperparameter settings would help. We will discuss the model and hyperparameter settings below. The main contribution of MTGRN is to infer GRNs through multivariate time-series prediction, where single-cell sequencing data is transformed into time-series data, and GRNs are inferred by predicting gene expression in the next M time points based on the gene expression in the previous N time points. MTGRN is composed of temporal attention module and spatial attention module. In the time module, each cell can only attend to cells from prior time points, allowing each gene to observe all genes in preceding cells. However, only a small subset of genes influences the expression of a given target gene. To address this, we use the spatial module to filter out gene pairs without regulatory relationships in the prior knowledge, ensuring that each gene’s expression is inferred based only on historical expression data of genes with known interactions in prior network. For hyperparameter settings, we applied grid search to optimize parameters, and the final hyperparameter values are as follows:
>   - **input_length** (number of cells in the previous N time points): 16
>   - **predict_length** (number of cells in the following M time points): 16
>   - **d_model** (dimension of Transformer): 128
>   - **d_ff** (dimension in feedforward): 512
>   - **heads** (number of attention heads in Transformer): 4
>
> Additional configurations, such as GPU setting, learning rate, and the number of training epochs, are described in detail in **Section 4** under **Reproducibility** paragraph, reviewer could refer to this section for more information. We hope our response solve your questions.
>
> 4. In response to reviewer's comments that we should incorporate dynamic gene expression data to infer dynamic networks. In fact, MTGRN is able to infer dynamic GRN, as we assign each cell a pseudotime and use time-series prediction to infer the GRN. During inference, we can segment cells along the differentiation trajectory into different time segments (e.g., grouping every $n$ consecutive cells into one segment). By inputting cells from each segment into MTGRN, we can obtain the GRN for that specific time segment. For the entire differentiation trajectory, assuming we generate $m$ time segment, we can apply Gaussian smoothing to the regulatory edge scores across these $m$ GRNs to construct a dynamic GRN. This approach is similar to Dictys [7]. We did not include this experiment in the main text of our submission, but we have now uploaded the results as supplementary material. Reviewer could examine these results in supplementary material, hoping this clarifies your questions!
>
> **Due to the character limit, we will give the reference links in the next comment.**

---

> ### Author Response · Authors · 2024-11-14
>
> the reference links are listed below:
>
> [1] Pratapa A, Jalihal A P, Law J N, et al. Benchmarking algorithms for gene regulatory network inference from single-cell transcriptomic data[J]. Nature methods, 2020, 17(2): 147-154.
>
> [2] Nestorowa S, Hamey F K, Pijuan Sala B, et al. A single-cell resolution map of mouse hematopoietic stem and progenitor cell differentiation[J]. Blood, The Journal of the American Society of Hematology, 2016, 128(8): e20-e31.
>
> [3] Hayashi T, Ozaki H, Sasagawa Y, et al. Single-cell full-length total RNA sequencing uncovers dynamics of recursive splicing and enhancer RNAs[J]. Nature communications, 2018, 9(1): 619.
>
> [4] Camp J G, Sekine K, Gerber T, et al. Multilineage communication regulates human liver bud development from pluripotency[J]. Nature, 2017, 546(7659): 533-538.
>
> [5] Wang P, Wen X, Li H, et al. Deciphering driver regulators of cell fate decisions from single-cell transcriptomics data with CEFCON[J]. Nature Communications, 2023, 14(1): 8459.
>
> [6] Kamimoto K, Stringa B, Hoffmann C M, et al. Dissecting cell identity via network inference and in silico gene perturbation[J]. Nature, 2023, 614(7949): 742-751.
>
> [7] Wang L, Trasanidis N, Wu T, et al. Dictys: dynamic gene regulatory network dissects developmental continuum with single-cell multiomics[J]. Nature Methods, 2023, 20(9): 1368-1378.
>
> We are very grateful for your valuable suggestions and hope our responses clarify any concerns. We greatly value this opportunity to discuss our work. If there are any further questions, please feel free to ask, and we will be happy to provide further details. We hope this discussion will help improve the score of our paper, and thank you again!

---

> ### Author Response · Authors · 2024-11-25
> **Anticipation of your response.**
>
> We greatly value the feedback provided by reviewers and sincerely apologize for any lack of clarity in presenting our work during the initial submission, which may have caused confusion. In the rebuttal phase, we have addressed the issues you raised in detail and the additional experiments you mentioned have been included in the supplementary material for your reference. We hope these clarifications and supplementary analyses will help you evaluate the significance of our study comprehensively.
>
> We are pleased to note that during the rebuttal phase, we successfully addressed the concerns of other reviewers, which led to a positive reevaluation and corresponding score improvements. We genuinely hope our detailed and thoughtful response will also resolve your concerns and help improve the score of our paper. We sincerely look forward to your feedback and thank you again!

---

> ### Author Response · Authors · 2024-11-25
> **Anticipation of your response**
>
> We greatly value the feedback provided by reviewers and sincerely apologize for any lack of clarity in presenting our work during the initial submission, which may have caused confusion. In the rebuttal phase, we have addressed the issues you raised in detail and the additional experiments you mentioned have been included in the supplementary material for your reference. We hope these clarifications and supplementary analyses will help you evaluate the significance of our study comprehensively.
>
> We are pleased to note that during the rebuttal phase, we successfully addressed the concerns of other reviewers, which led to a positive reevaluation and corresponding score improvements. We genuinely hope our detailed and thoughtful response will also resolve your concerns and help improve the score of our paper. We sincerely look forward to your feedback and thank you again!

---

> ### Author Response · Authors · 2024-11-28
> **Anticipation of your response**
>
> We are looking forward to receive feedback from the reviewer G2id.

---

> > ### Comment · Reviewer_G2id · 2024-11-28
> >
> > I appreciate the sincere effort the authors put in to address comments from all reviewers. I have raised my score to 6.

---

### Official Review · Reviewer_e1zu · 2024-11-04

**Soundness:** 2
**Presentation:** 3
**Contribution:** 2
**Rating:** 6
**Confidence:** 5

**Summary:**

The authors propose to predict gene regulatory network connections from scRNA-seq data by learning an attention matrix that captures weighted edges between genes. Pseudotime and prior knowledge are used to help the model learn the GRN. The authors show that the model beats previously published methods on the same task.

**Strengths:**

The proposed deep learning architecture for learning GRNs (attention matrix) and the interpretability methods the authors implement to identify key transcription factors are very interesting.

The paper is generally well-written and easy to understand.

**Weaknesses:**

The methods uses prior knowledge ("a highly comprehensive gene interaction network proposed in NicheNet") in the training phase and subsequently evaluates on "the ground truth network provided in Pratapa et al. (2020)". It is possible that the prior knowledge network and the evaluation network share information and this possible circularity was not tested. The potential (and maybe likely) circularity seriously undermines the performance evaluations.

The perturbation analysis is interesting, but this could be a separate paper by itself (e.g. with comparisons to other perturbation prediction methods). I would have liked to have seen a more thorough technical analysis of the main method, such as ablation studies, instead of a small add on showing the additional perturbation use case without much technical exploration.

**Questions:**

Can the authors show that the performance gains are not due to circularity between the prior and evaluation data?

Assuming no circularity, what are the technical aspects of the model architecture that contribute most to the performance? i.e. what aspects should others try to build on?

---

> ### Author Response · Authors · 2024-11-13
>
> We sincerely appreciate the reviewer’s thoughtful comments. we will address each of the questions below.
>
> 1. Regarding the question of whether prior knowledge plays a significant role in our model's performance, actually we proposed a method named **Random** in Table 1 for baseline comparison. In **Random**, we randomly selected the top-k edges (where k is equal to the number of edges in the ground truth) from the prior knowledge as the inferred GRN. As for results, we can find that the AUROC for **Random** across five datasets is around 0.5, significantly lower than that of our model (MTGRN), indicating that prior knowledge alone is not the reason for our model’s high performance.
> 2. Methods we compared against in Table 1 also rely on prior knowledge to infer GRNs. For example, CellOracle [1] uses ATAC-seq data to construct a base GRN, providing prior knowledge of TF-target relationships. CEFCON [2] also relies on NicheNet as prior information, applying a graph neural network to infer the GRN. For fairness, the baselines we chose also incorporate prior knowledge, and our model achieved higher performance in these comparisons, further supporting that our model’s performance is not due to the use of prior knowledge.
> 3. We analyzed the overlap between the edges provided by NicheNet and the edges in ground truth. The result is presented below:
>    - Prior Knowledge: 5318181 edges
>    - mESC ground truth network: 24557 edges (overlap: 16695)
>    - mHSC-E ground truth network: 24726 edges (overlap: 14732)
>    - mHSC-GM ground truth network: 16198 edges (overlap: 9818)
>    - mHSC-L ground truth network: 4705 edges (overlap: 2494)
>
>     we can find that NicheNet includes 5318181 edges, while the ground truth edge counts in our datasets are 24557, 24726, 16198, and 4705, with intersections of 16695, 14732, 9818, and 2494 edges, respectively. In each dataset, the effective regulatory edges provided by NicheNet account for only 0.31%, 0.27%, 0.18%, and 0.04% of the entire prior knowledge. This indicates that NicheNet provides a coarse prior network containing considerable noise, including potential false-positive regulatory relationships from databases or experiments. This is also why the AUROC for random is around 0.5 in Table 1. MTGRN should identify approximately 10,000 true regulatory edges from a noisy network of over 5 million edges, making this task challenging and countering the concern that our high performance is due to the strength of prior knowledge (actually the information provided by prior is little).
>
> 4. In response to the reviewer's question that what are the technical aspects of the model architecture that contribute most to the performance? we emphasize that MTGRN’s contribution is reframing GRN inference as a multivariate time series forecasting problem, learning Granger causality [3] in cell development. Our model’s ability stems from effectively utilizing the temporal information inherent in cell development. As shown in Figure 2, we first apply a ***Temporal Attention Module*** that allows each cell to focus on gene expression information from earlier developmental stages. At this module, each gene of the cell will attend the expression of all genes. However, each gene is typically influenced by only a subset of regulatory factors. We therefore apply the ***Spatial Attention Module***, which leverages prior knowledge to filter out gene pairs without regulatory relationships (although this filtering is somewhat coarse due to noise in the prior knowledge). Combining these two modules allows us using the historical expression information of candidate TFs to predict a gene’s future expression. This is the core factor improving MTGRN’s performance.
> 5. In response to the reviewer's question that what aspects should others try to build on? we believe building GRNs using methods related to RNA velocity or single-cell trajectory inference could yield additional insights. Incorporating other modalities, such as ATAC-seq data like CellOracle, could also strengthen the accuracy of prior knowledge, which avoids the noise present in NicheNet.
>
> We are very grateful for your valuable suggestions and hope our responses clarify any concerns. We greatly value this opportunity to discuss our work. If there are any further questions, please feel free to ask, and we will be happy to provide further details. We hope this discussion will help improve the score of our paper, and thank you again!
>
> [1] Wang P, Wen X, Li H, et al. Deciphering driver regulators of cell fate decisions from single-cell transcriptomics data with CEFCON[J]. Nature Communications, 2023, 14(1): 8459.
>
> [2] Kamimoto K, Stringa B, Hoffmann C M, et al. Dissecting cell identity via network inference and in silico gene perturbation[J]. Nature, 2023, 614(7949): 742-751.
>
> [3] Shojaie A, Fox E B. Granger causality: A review and recent advances[J]. Annual Review of Statistics and Its Application, 2022, 9(1): 289-319.

---

> ### Author Response · Authors · 2024-11-14
>
> We would like to clarify the perturbation experiments mentioned by reviewer in weakness section. Actually, it serve as an important technical validation of the inferred gene regulatory network. Similar approaches can be found in studies like Celloracle [1]. In fact, without perturbation analysis, it would be difficult to validate the inferred network’s accuracy. It is through the perturbation of key factors that the robustness of the network can be definitively proven.
>
> In Figure 7 (a), our TF perturbation analysis is entirely based on the GRN inferred by our model (MTGRN). If the inferred network were incorrect, the calculated offset vectors for each cell would deviate in the wrong direction. However, by performing perturbations on the network constructed by MTGRN, we observed that knocking out Gata1 (a well-known transcription factor promoting erythroid development) causes the offset vectors of developing cells to shift in the opposite direction of the developmental trajectory. This strongly supports the accuracy of our model’s GRN inference and serves as a robust technical validation. It is an important technical discussion for analyzing model performance, and we do not fully agree with the reviewer’s perspective on this point. We hope to have further discussion.
>
> We greatly appreciate the opportunity for this rebuttal and hope our responses address the reviewer’s concerns. Thanks once again!
>
> [1] Kamimoto K, Stringa B, Hoffmann C M, et al. Dissecting cell identity via network inference and in silico gene perturbation[J]. Nature, 2023, 614(7949): 742-751.

---

> ### Author Response · Authors · 2024-11-24
> **Request for your response**
>
> We greatly appreciate the reviewers’ feedback and hope they respect others’ work with careful consideration. During the rebuttal phase, we value constructive and evidence-based discussions rather than dismissive or careless remarks. We have thoroughly addressed all the concerns you raised, and if you have further questions, we kindly request your response. Thank you!

---

> > ### Comment · Reviewer_e1zu · 2024-11-25
> >
> > Thanks for the responses to the comments from all the reviewers. The removal of the NicheNet prior from the pipeline in response to another reviewer is a good demonstration of the performance of the method without the possibility of circularity, which was my biggest concern. I have raised my score.

---

> > > ### Author Response · Authors · 2024-11-25
> > >
> > > We are very pleased to have resolved your concerns and are grateful for the reviewer’s decision to improve our score. We hope this article will also prove helpful to you.

---

### Meta-Review · Area_Chair_bXx8 · 2024-12-18

**Metareview:**

The paper introduces a novel method for inferring gene regulatory networks (GRNs) using a transformer-based model. MTGRN identifies three crucial genes associated with mouse embryonic stem cell development.

Strengths include the methodology of using single-cell data and incorporating temporal and spatial blocks. The performance appears to be better than other multimodal approaches. Finally, the ability to conduct perturbation experiments and model changes in cell identity adds practical value to the method.

Weaknesses focused on validation, novelty and experiments. There were concerns about the validation of the inferred GRNs, particularly the potential overlap between the prior knowledge and the ground truth networks. Some reviewers felt that the use of transformers for GRN inference had been explored before, limiting the novelty of the contribution. Ablation studies were suggested to provide more evidentiary support for the claims.

Overall, while the paper presents a promising approach with strong empirical performance, addressing the concerns about the use of prior knowledge, validation, and providing more technical details could strengthen the submission. There was support from the reviewers to pursue this line of research but at this time, the submission seems to be slightly below the bar for acceptance.

**Additional Comments On Reviewer Discussion:**

The reviewers and authors engaged in an extensive discussion. After the discussion period the reviewers engaged with the AE to summarize the comments and come to an agreement about the general recommendation for the paper. The authors raised concerns about the reviews which were considered thoroughly. Reviewing is a voluntary and unpaid activity; some reviewers are able to engage more than others due to other responsibilities. Furthermore, some reviewers are closer to the specific field of the paper while others may be more distant. The conference has mechanisms to allow for the program chairs to make informed recommendations based on input from all the reviews. I appreciate the engagement that the reviewers were able to provide the community.

---

### Decision · Program_Chairs · 2025-01-22

Reject